# The Impact of Cerebral Amyloid Angiopathy on Functional Outcome of Patients Affected by Spontaneous Intracerebral Hemorrhage Discharged from Intensive Inpatient Rehabilitation: A Cohort Study

**DOI:** 10.3390/diagnostics12102458

**Published:** 2022-10-11

**Authors:** Carmen Barbato, Piergiuseppe Liuzzi, Anna Maria Romoli, Francesca Draghi, Daniela Maccanti, Andrea Mannini, Claudio Macchi, Francesca Cecchi, Bahia Hakiki

**Affiliations:** 1IRCCS Fondazione Don Carlo Gnocchi ONLUS, Via di Scandicci 269, 50143 Florence, Italy; 2NEUROFARBA Department, Neuroscience Section, University of Florence, 50139 Florence, Italy; 3Scuola Superiore Sant’Anna, Istituto di BioRobotica, Pontedera, Viale Rinaldo Piaggio 34, 56025 Pisa, Italy; 4Department of Experimental and Clinical Medicine, University of Florence, 50134 Florence, Italy

**Keywords:** cerebral amyloid angiopathy, spontaneous intracerebral hemorrhage, rehabilitation outcomes

## Abstract

Background: Sporadic CAA is recognized as a major cause of sICH and sABI. Even if intensive rehabilitation is recommended to maximize functional recovery after sICH, no data are available on whether CAA may affect rehabilitation outcomes. In this observational prospective study, to explore the impact of CAA on rehabilitation results, functional outcomes after intensive rehabilitation have been compared between patients affected by sICH with and without a diagnosis of CAA. Methods: All adults affected by sABI due to sICH and admitted to the IRU of IRCCS-Don-Gnocchi-Foundation were consecutively enrolled for 12 months. Demographic and clinical data were recorded upon admission and discharge. Results: Among 102 sICH patients (age: 66 (IQR = 16), 53% female), 13% were diagnosed as probable/possible-CAA. TPO and functional assessment were comparable upon admission, but CAA patients were significantly older (*p* = 0.001). After a comparable LOS, CAA patients presented higher care burden (ERBI: *p* = 0.025), poorer functional recovery (FIM: *p* = 0.02) and lower levels of global independence (GOSE > 4: *p* = 0.03). In multivariate analysis, CAA was significantly correlated with a lower FIM (*p* = 0.019) and a lower likelihood of reaching GOS-E > 4, (*p* = 0.041) at discharge, independently from age. Conclusions: CAA seems to be independently associated with poorer rehabilitation outcomes, suggesting the importance of improving knowledge about CAA to better predict rehabilitation outcomes.

## 1. Introduction

Sporadic cerebral amyloid angiopathy (CAA) is a cerebral small vessel disease, initiated by amyloid beta-peptide deposition within small- to medium-sized blood vessels of the brain and leptomeninges. Resultant vascular changes, such as concentric splitting of the vascular wall, microaneurysm formation, chronic inflammatory infiltrates, and fibrinoid necrosis, mainly trigger ischemic or hemorrhagic brain manifestations [1,2,3]. The incidence of CAA is strongly age dependent [4]. By autopsy, the prevalence of CAA ranged from 2.3% and 12.1% in patients over the age of 65 and it is higher in older patients with dementia [5]. As recommended by the modified Boston Criteria (mBC) [6,7,8], definite CAA is only diagnosed postmortem, while a diagnosis of CAA should be clinically suspected in patients aged 55 years or older, with or without a clinical manifestation of CAA, who have acute or chronic hemorrhagic findings on brain magnetic resonance imaging (MRI) in the absence of an obvious alternative cause. Analysis of biopsied brain tissue further may support the diagnosis, defined as probable CAA with supported pathology, but is uncommonly performed.

Clinically, CAA may present with several neurological manifestations, such as transient neurological symptoms, progressive cognitive impairment, or incidental leukoencephalopathy, microbleeds or cortical hemosiderosis on MRI. Therefore, the most common and devastating clinical manifestation of CAA is acute spontaneous intracerebral hemorrhage (sICH). The typical lobar location with subarachnoid extension of the hemorrhages reflects the underlying distribution of amyloid deposits and clinical presentation varies with the lesion size and brain region involved [9]. A high number of survivors of a CAA-related sICH often face severe acquired brain injuries (sABI), characterized by alteration in consciousness, sensorial, motor, cognitive, or behavioral impairment [10] and possibly permanent disability. Although, after the acute phase, most CAA-related sICH survivors frequently need intensive rehabilitation, which is recommended to maximize functional recovery [11,12,13]. SABI are defined as traumatic, post-anoxic, vascular, or other brain damages that cause an alteration in consciousness for at least 24 h, a Glasgow Coma Scale score that ranges between 3 and 8 after 24 h, and often lead to a permanent disability. The incidence rate of sABI has been estimated around 15 per 100,000 persons and cerebrovascular events are the second leading cause after traumatic brain injury. The sICH incidence represents about 28% of all cerebrovascular events [14]. However, the accurate impact of sICH on sABI of hemorrhagic etiology is not well studied and limited data are available about the explicit prevalence of sICH within sABI patients, whereas no data are available about CAA-related sICH [15]. Concerning prognosis after sICH, early clinical investigations have traditionally focused on mortality. Several studies have reported that the 30-day mortality rate from a sICH ranged from 32 to 52% [16]; one-half of these deaths occurred within the first two days. In population-based cohorts of patients hospitalized after sICH, the 10-year survival rate ranged from 18 to 25% [17] and life expectancy was decreased compared to the general population [18]. It is noteworthy that patients with lobar sICH, most commonly associated with CAA, have high mortality from 10 to 30% and a higher rate of recurrence if compared with non-CAA sICH patients [19,20]. Although mortality may be considered as an important indicator of disease severity, recently, more attention has been paid to disability outcomes in survivors [13,21]. Unexpectedly, few clinical studies have addressed the rehabilitation outcomes of sICH patients and, in the absence of high-quality clinical data to guide practice, rehabilitation of these patients is largely based on general principles learned from ischemic stroke recovery, even if some differences have been reported [22]. This lack could be explained by their ‘disappearance’ behind the diagnosis of sABI upon admission to the rehabilitation setting. As a consequence, specific clinical characteristics of sICH are usually lost because of such mischaracterization and generalization. This leads to an important gap from a rehabilitation point of view because one of the cornerstones of a successful rehabilitation pathway is the precise personalization of treatment, which implies good knowledge of the peculiarities of the patients. To date, no data are available regarding rehabilitation of CAA-related sICH survivors, neglecting an important key point that may help to plan a personalized rehabilitation pathway and to define an accurate long-term prognosis. 

In this context, we hypothesized that the peculiar CAA clinical and neuroradiological manifestations and the high rate of hemorrhagic recurrence, in addition to the delicate management required to avoid complications, may negatively influence the functional recovery during rehabilitation, if compared to nCAA-sICH. The primary aim of this study was to investigate the functional outcomes in patients with sABI caused by CAA-related sICH, compared to a reference group of non-CAA sICH, and, as a secondary aim, to evaluate the prevalence of CAA upon intensive rehabilitation unit (IRU) admission.

## 2. Materials and Methods

This was a single cohort prospective observational study and data were collected using the framework of a multicenter prospective observational longitudinal study that explores the clinical, genetical and neurophysiological predictors of functional recovery in sABI that involves four IRUs of the Fondazione Don Carlo Gnocchi Institute in Italy [23]. Ethical approval was obtained by the local Committee and all patients or their principal caregivers signed an informed consent to participate (N. 16606_OSS). The study was registered on ClinicalTrials.gov with the following registration number: NCT04495192. The present sub-analysis was conducted following STROBE guidelines [24] (Figure 1).

### 2.1. Participants 

All patients that came from an IRU with a diagnosis of sABI due to a sICH, with at least one brain MRI performed after the sICH onset, aged ≥ 18 years and admitted to IRU of IRCCS-Don Carlo Gnocchi Foundation (Florence, Italy) from June 2020 to June 2021 were recruited. All patients who had a diagnosis of non-spontaneous (traumatic) ICH were excluded. No additional exclusion criteria, other than the absence of a signed informed consent or age out of range, were considered. 

### 2.2. Clinical and Instrumental Assessment

At admission, demographic and clinical data were recorded, including those related to the acute event of sICH. In particular, sICH etiology (hypertensive hemorrhages, CAA-related hemorrhage, related arteriovenous and other vascular malformations rupture hemorrhage, hemorrhagic infarction, etc.), time post onset (TPO) and occurrence of clinical complications were recorded. The presence of sepsis or epileptic crisis during the IRU stay was also recorded.

Within the first week after admission, an extensive clinical and instrumental evaluation was performed by a team of skilled professionals (including neurologists, speech therapists, and physiotherapists) and included the following: (1) the consciousness state using the Coma Recovery Scale-Revised (CRS-R) [25]; (2) the care burden using the Early Rehabilitation Barthel Index (ERBI) [26]; (3) the functional disability assessed by the Functional Independence Measure (FIM) [27]; (4) the autonomy level using the Glasgow Outcome Scale-Expanded (GOS-E) [28]; (5) the neurocognitive and behavioral assessment using the Level of Cognitive Functioning scale (LCF) [29]. Additionally, a complete neurophysiological evaluation including (1) a standard electroencephalography (EEG), using the American Clinical Neurophysiology Society’s Standardized Critical Care EEG Terminology classification [30,31], and an electromyography to assess the presence of critical polyneuromyopathy [32] were also performed. Full clinical assessment was repeated at discharge.

### 2.3. Diagnosis of CAA-Related sICH 

Since the patients were not admitted with an etiologic definition of their hemorrhage, they were all reviewed by a neurologist trained and experienced in CAA. According to the mBC [6,7,8], the main diagnostic categories for clinical practice and research (<<probable CAA>> or <<possible CAA>>) were considered for our purpose. 

As formulated in mBC, CAA is suspected in patients aged 55 or older with a sICH, in the absence of an obvious alternative cause. <<Probable CAA>> required neuroimaging demonstration of multiple hemorrhages restricted to lobar brain regions, cortico-subcortical junction and subcortical white matter or a singular hemorrhage in the above-mentioned regions, plus cortical superficial siderosis (cSS). Presence of just one hemorrhagic manifestation (hemorrhage or cSS) identified a <<possible CAA>>. Additionally, for the definition of a <<possible CAA>>, we required at least one other CAA-related white matter lesion (CSO-EPS: enlarged perivascular spaces of the centro semiovale or WMH-MS: with matter hyperintensity with a multisport pattern) to improve the diagnostic accuracy. 

Because CAA typically spares deep territories, the presence of hemorrhagic lesions in basal ganglia, thalamus, or pons precluded the probable or possible CAA diagnosis. Based on patients’ clinical history and available neuroimaging examinations, the <<probable CAA>> and <<possible CAA>> diagnoses were reported and all misdiagnosis rectified. 

### 2.4. Rehabilitation Treatment

During the IRU stay, all patients received a multi-professional interdisciplinary rehabilitation treatment. The Individual Rehabilitation Project (IRP) was based on the patient’s cognitive level of functioning and clinical necessities upon admission. The treatment consists of an average of 3 h of specific treatment per day delivered by an interdisciplinary team of professionals, including physiotherapy, rehabilitation nursing management, and speech and language therapy, occupational therapy, neuropsychological assessment and treatment and psychological support to patients and families according to the emerging needs. 

Discharge was planned with family and caregivers and carried out upon decision of an interdisciplinary team, in agreement with the local Health Authority, either when the patient reached a plateau, or when the patient achieved a functional improvement that allowed home discharge or transfer to a less specialized intensive rehabilitation setting.

### 2.5. Pharmacological Treatment

According to our clinical protocol, pharmacological interventions were planned that agreed with the patient’s emerging needs. Particularly in all our patients affected by sICH, the main objective was to prevent hemorrhage extension and other neurologic and medical complications (prevention of aspiration, venous thromboembolism, pressure-induced skin injury, fever, hyperglycemia and hypoglycemia management, etc.). All anticoagulant and antiplatelet drugs were discontinued and we managed elevated blood pressure as recommended by the guidelines from the American Heart Association [33].

### 2.6. Outcomes

The primary outcome was the achievement of a moderate functional autonomy (GOS-E > 4) at discharge. Secondary outcomes were as follows: (1) the improvement of consciousness/cognitive state (CRS-R and LCF), (2) the improvement of the functional disability (FIM) and (3) the reduction in the care burden weight (ERBI) at discharge.

### 2.7. Statistical Analysis

Descriptive analyses and group comparisons were carried out between admission variables and the presence/absence of CAA. In particular, numerical independent variables underwent logistic regression and categorical independent variables underwent a chi-square analysis. Then, the outcomes at discharge were compared between patients with (CAA) and without (nCAA). Numerical outcomes (CRS, LCF, FIM and ERBI) were analyzed using either a t-test or a Mann–Whitney test conditioned to their significance to a normality test (Shapiro–Wilk), with the grouping variable set to the presence of CAA. Categorical outcome (GOS-E > 4) underwent a chi-square analysis with the presence of CAA as the dependent variable. 

Admission and discharge variables were compared for the CAA group to evaluate whether a significant improvement was obtained during the rehabilitation stay (via either paired samples t-tests or Wilcoxon sum rank tests, conditioned to normality results).

Lastly, multivariate regressions (logistic for the categorical outcome and linear for the numerical ones) were conducted for each significantly different scale as the dependent variables together with age, gender, TPO, LoS and diagnosis of CAA as the independent variables.

## 3. Results

One hundred and two patients affected by sICH were enrolled in the study (54 (52.9%) women, median age 66 years (IQR = 16), the median TPO was 64 days (IQR = 27)). Demographic and clinical characteristics are summarized in Table 1. 

Based on patients’ clinical history and neuroimaging examinations, a total of thirteen (12.7%) patients had a clinical diagnosis of CAA and in particular, seven (6.9%) were diagnosed as probable CAA and six (5.9%) as possible CAA. Within the non-CAA (nCAA) group, 60 patients suffered from a primary or hypertensive intraparenchymal hemorrhage, 16 from a sICH caused by a ruptured aneurysm, and 13 from a sICH caused by an arteriovenous malformation.

Except for the older age in the CAA group (median nCAA = 64 (IQR = 14) and CAA = 76 (IQR = 12), *p* = 0.005), no significant differences were found in the admission variables related to the presence or not of CAA regarding the clinical and functional evaluations, electroencephalographic and electromyographic characteristics, and the presence of medical devices upon admission (Table 1).

At discharge, after a comparable length of stay (LOS) of 87 days (IQR = 75) for CAA patients vs. 84 days (66) for nCAA patients; *p* = 0.52, the CAA group showed significantly lower scores in the included functional scales (Table 2). The median ERBI in the CAA group was −255 (IQR = 176) versus −160 (IQR = 113) in the nCAA, (*p* = 0.025) and the median FIM score was 18 (IQR = 3) in the CAA group versus 24.5 (IQR = 19) in the nCAA group (*p* = 0.02). A moderate functional autonomy (GOS-E > 4) was achieved in 15.9% of the CAA group compared to 46.9% of the nCAA group (*p* = 0.032). No significant differences were found for the consciousness/cognitive outcomes (CRS-R: *p* = 0.092; LCF: *p* = 0.081).

To address whether CAA patients had an improvement during the IRU stay, a paired comparison between admission and discharge of the outcomes scores was performed using the T-tests or Wilcoxon rank sum test for numerical variables, while McNemar chi-square tests were used for categorical variables. Consciousness levels (CRS-R, *p* = 0.021) and rehabilitation care burden (ERBI, *p* = 0.011) were found to be significantly different within the CAA group (Table 3).

Considering that the two groups (CAA and n-CAA) had a significantly different scores at discharge on GOS-E, FIM and ERBI (Table 2), further multivariate analysis (for each scale as dependent variable) were performed including as independent variable: CAA, sex, age, TPO, LOS and the value of the target scale itself in its assessment upon admission. In the multivariate step, the presence of CAA was significantly correlated with a lower FIM at discharge (*p* = 0.019, *β* = −14.627, 95%CI = −26.790/−2.464), together with a lower FIM score upon admission (*p* < 0.001), independently from age, gender, time post-onset and length of stay. CAA was also significantly associated with a higher probability to achieve good autonomy (GOS-E > 4) at discharge, (*p* = 0.041, OR = 0.232, 95%C.I. for EXP (*β*) = 0.134–0.901), independently from age. CAA was not associated with the ERBI (*p* = 0.091) whilst the ERBI upon admission influenced the related discharge value (*p* < 0.001). (Table 4).

## 4. Discussion

Although CAA is well recognized as a major cause of sICH, with a high mortality and disability risk, it continues to be largely misrecognized in rehabilitation settings and its possible interference with the rehabilitation course underestimated. The present study investigated the prevalence and the influence of CAA in sABI patients from a rehabilitative perspective. In our cohort, while showing improvement in all functional assessment scales during their stay at the IRU, patients with sICH affected by CAA had worse outcomes after a comparable length of stay, when compared with the nCAA group.

After a median length of stay of 87 days in our IRU and an intensive rehabilitation program, CAA patients showed a significant improvement of their consciousness state and their care burden between admission and discharge. However, only 15.8% of the CAA group reached moderate functional autonomy (GOS-E > 4) in comparison with 46.9% of the nCAA group, even after a comparable LOS in the IRU. In addition, the achievement of some rehabilitation milestones, such as reaching a higher functional ability and autonomy, seems to be significantly penalized by the presence of CAA. Indeed, in the multivariate analysis, the presence of CAA (independently from the older age) was significantly associated with lower outcomes (FIM and GOS-E). These findings may be explained by the fact that CAA-related sICH is characterized by a global progressive cerebral microangiopathy inducing several neurological manifestations [3,4,5,6]. The amyloid beta-peptide deposition within the cerebral vasculature in CAA patients is a progressive phenomenon that causes several clinical features, even before the index event of the sICH [1,2,3]. Pathological changes in the small vessels can lead to chronic manifestations beyond the hyperacute presentation. In clinical practice, the majority of patients diagnosed with CAA, either pre- and post- sICH, appear to have full developed or hidden cognitive impairment, which may reflect the comorbid Alzheimer’s disease, vascular dementia, or both [34,35]. Likewise, convexity subarachnoid hemorrhage and cerebral microbleeds can also occur either remotely from acute sICH or in the absence of sICH. These lesions may be associated with positive or negative focal neurologic symptoms and behavioral signs [36]. Against the scientific background in which few clinical studies have addressed rehabilitation outcomes of sICH patients and no data are available regarding the rehabilitation of CAA-related sICH survivors, our results emphasize that the peculiarity of CAA should be considered in the IRP, to decide its intensity and duration and to provide correct information about prognosis and expected functional recovery to the caregivers. 

Moreover, better knowledge of CAA in rehabilitation settings is required to improve the pharmacological management of these patients. Indeed, survivors of lobar hemorrhage and patients with other clinical manifestations of CAA are at high risk for future hemorrhagic complications. This risk should be factored into decision-making when assessing the risks and benefits of other medications. In this scenario, risks and benefits of using anticoagulant [37,38] and antiplatelet agents [39] may be carefully weighed and cardiovascular risk factors should be closely controlled to avoid future complications. Moreover, several drugs that interact with coagulation functioning and are commonly used in the management of hemorrhagic sABI complications should be avoided.

Upon admission, no significant differences were found between the CAA and the nCAA group regarding the clinical, functional evaluations and neurophysiological markers, upon admission. However, patients affected by CAA were significantly older than non-CAA patients, corroborating the well-known notion that incidence of CAA is strongly age dependent [40,41]. Specifically, by autopsy, the prevalence of CAA ranged from 2.3% for patients between the ages of 65 and 74 to 12.1% in patients over the age of 85 and even higher in older patients with dementia [4,5]. Nevertheless, the limitation due to the small sample size might have led to an underestimation of some differences between the two groups that cannot be ignored and further studies on a larger cohort should be performed to better address this issue. 

The secondary aim of this work was to estimate the CAA prevalence in an intensive rehabilitation setting. In a total of 102 patients affected by sICH and admitted to our IRU, the prevalence of CAA was around 13%. This finding is in line with the finding reported in the literature for patients with a similar median age to our sample [40,42]. It should be noted, however, that in the absence of an etiological sICH diagnosis upon admission, the CAA diagnoses were derived from a careful clinical review of all patients admitted with a sICH. We decided to consider both diagnostic categories (possible and probable) to define our CAA patients. In particular, in the definition of a <<possible CAA>> diagnosis, we took into account the presence of at least one other CAA-related white matter lesion (CSO-PVS or WMH-MS). As suggested by the new Boston Criteria v2.0 [43], this MRI finding, together with the hemorrhagic marker of a single sICH, in the absence of any deep hemorrhagic lesions and other cause of hemorrhagic lesions, was defined as a <<probable CAA>>, improving sensitivity and providing overall superior diagnostic accuracy [43]. Numerous rectifications of misdiagnosis have been required to calculate a reliable prevalence. The high number of misdiagnoses might be due to the specialized ability required to make a proper diagnosis [40,41] and also to the mischaracterization of these patients, who generically are considered as common sABI due to sICH. Additionally, there still exists the incorrect assumption that patients affected by CAA can be considered and treated as common sICH, and are consequently misdiagnosed [22]. 

The present study has some limitations that should be discussed. First, this study was conducted in a single center with a small sample size of the patients admitted. Thus, the generalization of our results should be carried out with caution, even if we consider that our patients may represent the middle band of all hemorrhagic patients and if we take into account the global prevalence of CAA in the general population [42]. Second, this study focused on rehabilitation outcomes at discharge, while a long-term follow-up would be necessary to better define the rehabilitation prognosis. Otherwise, this study was a prospective cohort study performed within a longitudinal multicenter observational cohort study, with a planned follow-up period of 24 months. Against this background, our objectives will be to indagate long-term functional outcomes of CAA-related sICH in a larger and multicenter cohort of sICH patients with a long-term follow-up period. 

## 5. Conclusions

To the best of our knowledge, this is the first study that assessed the CAA specificity in a rehabilitation setting by exploring its impact on rehabilitation outcomes and its prevalence. Our results suggest a significant negative effect of CAA on rehabilitation efficacy in patients with sICH. Although the patients with CAA-related sICH are admitted to intensive rehabilitation departments that gather all the sABI patients, clinicians must have high specialized competences to identify the peculiar clinical and neuroradiological manifestations of this pathology. Making an accurate diagnosis of CAA and identifying the specific cause of sICH provide important evidence about the prognosis of sABI patients and it guarantees proper clinical and pharmacological management, avoiding inappropriate mistakes in daily decision-making. In the absence of previous data, these findings should be confirmed in large sample studies. Further data may be determinant to improve functional prognosis and pharmacological management of these peculiar subgroup of patients.

## Figures and Tables

**Figure 1 diagnostics-12-02458-f001:**
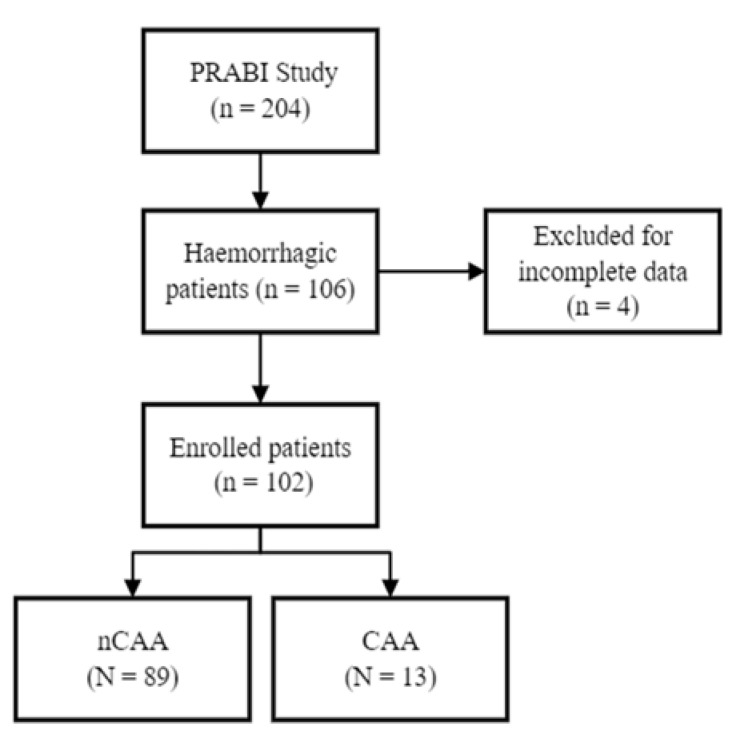
Enrollment flowchart. Legend: PRABI study: clinical, neurophysiological and genetic, predictors of recovery in patients with severe acquired brain injuries; CAA: cerebral amyloid angiopathy; nCAA: non-cerebral amyloid angiopathy.

**Table 1 diagnostics-12-02458-t001:** Characteristics of the study sample upon admission.

	Total Sample (N = 102)	nCAA (N = 89)	CAA (N = 13)	OR/*X*^2^	95%CI	*p*-Value
Clinical and Functional Evaluations						
Age, years	66 {16}	64 {14}	76 {12}	1.109	1.032–1.191	0.005
Gender, F	54 (52.9)	48 (53.9)	6 (46.2)	0.275	--	0.768
TPO, days	40 {27}	40.5 {24}	36 {51}	0.996	0.977–1.015	0.659
CRS-R total score	17.5 {14}	16 {14}	19 {14}	0.995	0.914–1.084	0.916
ERBI score	−275 {0}	−275 {10}	−275 {0}	0.990	0.970–1.012	0.379
FIM	18 {2}	18 {2}	18 {4}	1.022	0.936–1.116	0.627
LCF scale	3 {2}	3 {1}	3 {2}	0.830	0.448–1.536	0.552
GOS-E	3 {0}	3 {0}	3 {1}	0.958	0.824–1.114	0.579
EEG						
Symmetry	43 (42.6)	36 (40.9)	7 (53.8)	0.775	--	0.549
Frequency				0.000	--	1.000
Alpha	34 (33.7)	30 (34.1)	4 (30.8)	--	--	--
Theta	67 (66.3)	58 (65.9)	9 (69.2)	--	--	--
Gradient AP				2.450	--	0.367
Absent	14 (13.9)	11 (12.5)	3 (23.1)	--	--	--
Present	71 (70.3)	64 (72.7)	7 (53.8)	--	--	--
N/A	16 (15.8)	13 (14.8)	3 (23.1)	--	--	--
Reactivity				5.283	--	0.121
Present	9 (8.9)	6 (6.8)	3 (23.1)	--	--	--
Not Constant	32 (31.7)	28 (31.8)	4 (30.8)	--	--	--
Not Clear	51 (50.5)	47 (53.4)	4 (30.8)	--	--	--
Absent	9 (8.9)	7 (8.0)	2 (15.4)	--	--	--
Voltage				3.689	--	0.089
Normal	92 (91.1)	82 (93.2)	10 (76.9)	--	--	--
Low-Voltage	9 (8.9)	6 (6.8)	3 (23.1)	--	--	--
Continuity				3.468	--	0.505
Continous	96 (95)	84 (95.5)	12 (92.3)	--	--	--
Quasi-continous	2 (2)	1 (1.1)	1 (7.7)	--	--	--
Discontinous	2 (2)	2 (2.3)	0 (0)	--	--	--
Burst-suppression	1 (1)	1(1.1)	0 (0)	--	--	--
Epileptic graphoelem	20 (19.8)	17 (19.3)	3 (23.1)	--	--	0.718
EMG						
CIPNM presence	58 (62.4)	52 (63.4)	6 (54.5)	0.325	--	0.742

Numerical variables are described via median and interquartile range (in brackets) and categorical independent variables as count and percentages (in parenthesis). Logistic regressions were performed for numerical independent variables and chi-square analysis for categorical ones. Odds ratio (OR) and 95% confidence intervals (95%CI) refer to logistic regressions and ***X***^2^ refers to chi-square tests. **Legend**: TPO: time post-onset; CRS-R: Coma Recovery Scale-Revised; ERBI: Early Rehabilitation Barthel Index; FIM: Functional Independence Measure; LCF: level of cognitive function; GOS-E: Glasgow Outcome Scale-Extended; EEG: electroencephalography, EMG: electromyography; CIPNM: critical illness polyneuropathy and myopathy.

**Table 2 diagnostics-12-02458-t002:** Demographic and clinical characteristics of the study sample at discharge.

	Total Sample (N = 102)	nCAA (N = 89)	CAA (N = 13)	Test Statistics	*p*-Value
GOS-E > 4	32 (41.6)	30 (46.9)	2 (15.4)	5.621	0.032
CRS-R total score	23 {0}	23 {0}	23 {4}	243.5	0.607
LCF	5 {2}	5 {1}	3 {1}	133.5	0.081
FIM	24 {17}	24.5 {19}	18 {3}	98.5	0.020
ERBI	−165 {113}	−160 {113}	−225 {176}	155.5	0.025
LOS, days	83 {68}	84 {66}	87 {75}	350.5	0.525
Sepsis during IRU stay	22 (21.8)	18 (20.5)	4 (30.8)	0.659	0.472
Epileptic seizures during IRU stay	5 (5.6)	5 (4.9)	0 (0)	--	1.000

Numerical variables are described via median and interquartile range (in curly brackets) and categorical independent variables as count and percentages (in parenthesis). Logistic regressions were performed for numerical independent variables and chi-square analysis for categorical ones. Odds ratio (OR) and 95% confidence intervals (95%CI) refer to logistic regressions and ***X***^2^ refers to chi-square tests. Legend: GOS-E: Glasgow Outcome Scale-Extended; CRS-R: Coma Recovery Scale-Revised; LCF: level of cognitive function; FIM: Functional Independence Measure; ERBI: Early Rehabilitation Barthel Index; LOS: length-of-stay.

**Table 3 diagnostics-12-02458-t003:** Admission–discharge comparisons.

	Z	*p*-Value
GOS-E > 4	−1.414	0.157
CRS-R total score	−2.210	0.021
LCF	−0.447	0.655
FIM	−0.024	0.994
ERBI	−3.145	0.011
GOS-E	−1.807	0.071

Legend: GOS-E: Glasgow Outcome Scale-Extended; CRS-R: Coma Recovery Scale-Revised; LCF: level of cognitive function; FIM: Functional Independence Measure; ERBI: Early Rehabilitation Barthel Index.

**Table 4 diagnostics-12-02458-t004:** Multivariate analysis: (A) GOS-E > 4; (B) FIM score; (C) ERBI score at discharge.

A: GOS-E ≥ 4 Nagelkerke R^2^ = 0.371	*p*-Value	Odds Ratio	95%C.I. for EXP(B)	
Age	0.220	0.969	0.923	1.019
Gender	0.078	0.371	0.123	1.117
CAA	0.041	0.232	0.134	0.901
TPO	0.528	0.994	0.977	1.012
LoS	0.389	0.995	0.983	1.007
GOS-E upon admission	0.022	5.256	1.271	21.735
**B: FIM** **R^2^ = 0.341**	** *p* ** **-Value**	**Odds Ratio**	**95%C.I. for EXP(B)**	** *p* ** **-Value**
Age	0.869	0.014	−0.152	0.179
Gender	0.072	5.268	4.429	17.253
CAA	0.019	−14.627	−26.790	−2.464
TPO	0.161	−0.092	−0.221	0.038
LoS	0.076	−0.068	−0.143	0.007
FIM upon admission	0.000	1.155	0.693	1.617
**C: ERBI** **R^2^ = 0.422**	** *p* ** **-Value**	**Odds Ratio**	**95%C.I. for EXP(B)**	** *p* ** **-Value**
Age	0.161	−0.809	−1.949	0.331
Gender	0.106	11.627	−6.037	21.291
CAA	0.091	−14.660	−18.366	9.045
TPO	0.087	−0.671	−1.441	0.100
LoS	0.345	−0.227	−0.702	0.249
**ERBI**	**0.039**	**0.372**	**0.019**	**0.724**

Legend: GOS-E: Glasgow Outcome Scale-Extended; TPO: time post-onset; LoS: length-of-Stay; CAA: cerebral amyloid angiopathy; FIM: Functional Independence Measure; ERBI: Early Rehabilitation Barthel Index.

## Data Availability

Data will be made available upon request to the corresponding author. The data are not publicly available due to still ongoing studies on the same dataset.

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
