# Peer review of "The Impact of Cerebral Amyloid Angiopathy on Functional Outcome of Patients Affected by Spontaneous Intracerebral Hemorrhage Discharged from Intensive Inpatient Rehabilitation: A Cohort Study"

_diagnostics, 2022, doi:10.3390/diagnostics12102458_

Round 1

Reviewer 1 Report

The strenghts of this article are the statistical analysis, well conducted, and the language.

However, there is a major concern with a dramatic impact on patients' stratification, thus critically influencing data interpretation. The problem is about the definition of CAA itself. It is well known that cerebral amyloidosis is often accompained by a severe involvement of the peripheral nervous system (PNS), characterized by moderate-to-severe entrapment syndromes. Moroever, patients with amyloidosis frequently suffer from hearing loss, cardiopathies and small fibers neuropathies. All these features, along with axial disturbaces and severe weight loss, have extensively described as major/minor diagnostic criteria for cerebral amyloidosis.

Finally, it would be interesting to differentiate patients with isolated CAA (do they really exist?) from those with either a predominant peripheral (PNS) or non-neurological phenotype.

In this connection, the Authors stated that an electromyographic assessment was conducted, in order to exclude a possible "critical illness neuromyopathy". Unfortunately neurophysiological data about EMG assessment were not presented (why?); moroever, the goal of such an evaluation should be the diagnosis of entrapment syndromes.

Another critical concern is about EEG assessment. It is not acceptable for a scientific paper to present data as they appear in Table#1 (e.g. "burst suppression", or "suppression burst", pattern is commonly accepted as a "discontinuous" pattern). When did the Authors conduct the EEG assessment? What about potentially interfering drugs? And sample traces?  A detailed description of periodic/epileptiform discharges is completely lacking.

Finally, the Authors concluded that patients with CAA have a poor prognosis, but this statment does not have any practical purpose.

Reviewer 2 Report

In this study, the authors report on the different outcomes after rehabilitating patients with ICH with and without CAA. The paper is well written. While reading the paper I came across several things that need further clarification:

1. The title is a bit misleading. In this retrospective study, the patients took into account not only patients with CAA and sICH, but also patients with sICH without CAA. In fact, the number of patients without ICH was much bigger (around 60) compared to the number of patients with CAA (n=13). Would it be possible to change the Title so it more precisely reflects the content of the paper? 

2. I am not sure if one could claim that the difference between groups was due to CAA only. For example, these patients were significantly older than the group of patients without CAA. Was there any difference in the way another medical treatment was conducted in these groups of patients? E.g. Rate of operative treatment, or rate of complications...

3. Multifactorial analysis is ok, but the number of patients with CAA (n=13) is to low so to draw definite conclusions on the effect of CAA on the outcomes after treatment.

4. The conclusions need to be changed. Is this really the best of the conclusions that could be drawn based on the results of the study?

5. These sentences are not clear to me "Otherwise, this study was performed in the context of a longitudinal multicenter observational cohort study with a planned follow-up of 24 months. In this context, our objectives will be to indagate long term functional outcomes of related-CAA sICH in a larger and multicenter cohort of sICH patients with a long-term follow-up." Was this study conducted as a part of another multicentric study? Please clarify! 

Reviewer 3 Report

Reading the manuscript written by Barbato et al., was really interesting. The work aimed to evaluate the prevalence of CAA at the IRU admission and to investigate the functional outcomes in patients with sABI caused by related-CAA sICH compared to a reference group of non-CAA sICH.

Overall, this paper is written in professional English with sufficient introduction, detailed methods and solid data. The article is easy to read, well designated and presented, and can be of interest to reader and researchers.

-       I still think that the article is not written in the journal format, please check and correct this;

-       Please be careful that there are words written twice next to each other in the same sentence (e.g: manifestation manifestations);

-       Please define all abbreviations before first use;

-       The discussion section should be enriched with much more data being also accompanied by several bibliographic references.

Round 2

Reviewer 1 Report

Some changes, especially regarding point#1, have been made, but I think that the paper still has the same limitations I've previously raised.

Reviewer 2 Report

No further comments.